# Effect of Probiotics on *Tenebrio molitor* Larval Development and Resistance against the Fungal Pathogen *Metarhizium brunneum*

**DOI:** 10.3390/insects13121114

**Published:** 2022-12-02

**Authors:** Sabina Dahal, Annette Bruun Jensen, Antoine Lecocq

**Affiliations:** Department for Plant and Environmental Sciences, University of Copenhagen, Thorvaldsensvej 40, 1871 Frederiksberg C, Denmark

**Keywords:** mealworm, production, immune response, probiotics, disease

## Abstract

**Simple Summary:**

Insects have been widely studied as a potential sustainable source of proteins to meet a rising global demand. Among them, the yellow mealworm (*Tenebrio molitor*, L.) is showing promise for its mass-rearing potential and its authorization by the European Union (EU) as a novel food. As in conventional animal husbandry practices, probiotics could provide nutritional and immunological benefits as part of the insect’s diet. This study evaluated the dietary supplementation of three types of probiotics on the development and disease resistance of yellow mealworm larvae. The results showed that the addition of probiotics can play a role in insect farming to improve the nutritional value of sub-optimal diets and protect the insects against entomopathogens. However, this study emphasizes the contrasting effects of the different probiotic strains tested and the need for more research on the topic.

**Abstract:**

In recent years, the yellow mealworm (*Tenebrio molitor* L.) has demonstrated its potential as a mass-produced edible insect for food and feed. However, challenges brought on by pathogens in intensive production systems are unavoidable and require the development of new solutions. One potential solution is the supplementation of probiotics in the insect’s diet to obtain the double benefits of improved growth and enhanced immune response. The aim of this study was to evaluate the effects of diet-based probiotic supplementation on *T. molitor* larval survival, growth, and resistance against a fungal pathogen. Three probiotic strains, namely *Pediococcus pentosacceus* KVL-B19-01 isolated from *T. molitor* and two commercialized strains for traditional livestock, *Enterococcus faecium* 669 and *Bacillus subtilis* 597, were tested. Additionally, when larvae were 9 weeks old, a pathogen challenge experiment was conducted with the fungus *Metarhizium brunneum*. Results showed that both *P. pentosaceus* and *E. faecium* improved larval growth and larval survival following fungal exposure compared to the non-supplemented control diet. Since *B. subtilis* did not improve larval performance in terms of either development or protection against *M. brunneum*, this study suggests the need for further research and evaluation of probiotic strains and their modes of action when considered as a supplement in *T. molitor*‘s diet.

## 1. Introduction

Though entomophagy is a common practice in many parts of the world [1], the commercial mass production of insects is a novel approach to addressing issues surrounding sustainable food systems [2] and nutrient recycling [3]. Insects are rich in protein, fatty acids, vitamins, and minerals [4]. Some species are well-suited for mass production as animal source food (ASF) substitutes and feed ingredients for livestock and pets [5]. One of the most promising insect species within the emerging insects-as-food-and-feed industry is the yellow mealworm (*Tenebrio molitor* L.) (Coleoptera: *Tenebrionidae*) [6]. Since this insect species can be easily managed for indoor rearing, it has been industrially produced as feed for pets and zoo animals such as wild birds, reptiles, small mammals, and so on [5]. Moreover, recent advancements in EU legislation have also paved the way for the use of *T. molitor* as a novel human food [7].

With growing concerns around replacing conventional protein sources with sustainable alternatives, more companies have engaged in scaling up insect production [8]. However, industrial mass production is dependent on gaining a better understanding of the insect’s biology and associated biological risk factors [9,10,11]. Risks associated with pathogen occurrence and disease transmission in mass insect production have been well-documented [10,12,13]. Entomopathogens include a range of bacteria, viruses, fungi, and nematodes [10,13]. Among them, fungal infections are a common problem in mass rearing systems, likely due to the humid conditions in rearing chambers favouring fungal growth [13,14,15]. One of the potential biological risk factors that can invade insect colonies is *Metarhizium brunneum*, a hypocrealean entomopathogenic fungus (EPF). Through the presence of adhesion factors, hydrolytic enzymes, and specialized infection structures, it is capable of infecting insects by penetrating the hard cuticle, leading to death within a few days [13,15]. Studies have shown this fungus to be highly virulent to Coleopteran insects, including *T. molitor* [12,13], and it is commonly used as a biological control agent against various pest species [12,15].

As with conventional production animals, the supplementation of viable beneficial microbes in the insect’s diet is a possible way to improve overall health status and prevent disease risks in the rearing system [9,14]. Strengthening the host immune system in insect farming could also provide protection against the emergence of opportunistic microbials when insects are reared for human food production [7,9,14]. These beneficial microbes, commonly known as probiotics, can confer health benefits to the host when consumed in adequate amounts [15,16,17,18,19]. Compared to probiotics for higher vertebrates, these supplemented microbes are gut commensals isolated and characterized from insects’ guts to obtain nutritional and health benefits [14]. The Most probiotic bacteria studied for insect feeding trials belong to a group of lactic acid-producing bacteria (LAB) such as *Lactobacillus* sp. and others such as *Saccharomyces* sp., *Streptococcus* sp., and *Bacillus* sp. [14]. In addition, recent studies of probiotic feeding in insect species such as *T. molitor* and *Galleria mellonella* have shown anti-microbial and anti-fungal properties against pathogenic bacteria such as *Bacillus thuringiensis*, *Serratia,* and *Pseudomonas* spp. in addition to fungal spp. such as *Candida albicans* [9,14]

The objective of this study was to evaluate the effect of probiotic-supplemented oatmeal diets on *T. molitor* larval growth and resistance against the fungal entomopathogen *M. brunneum*. Three probiotic strains were used in this study: two strains used as probiotics in conventional livestock, *B. subtilis* 597 and *Enterococcus faecium* 669, and a strain previously isolated from *T. molitor*, *Pediococcus pentosaceus* KVL-B19-01.

## 2. Materials and Methods

### 2.1. Experimental Insects

*Tenebrio molitor* larvae used in the present study were procured from adult beetles reared at the Section for Organismal Biology (SOBI) of the University of Copenhagen (UCPH), Denmark. Eight small Petri dishes (8.5 cm diameter and 1.4 cm high) filled with commercially available potato starch flour and covered with a mesh (average mesh size of 1.62 cm × 0.78 cm (Olympus provis, Miami, FL, USA)) were placed in cages with adult beetles. After 24 h, the eggs were collected by sieving the flour using a metal strainer. The collected eggs were incubated at 30 °C in three large Petri plates (13.5 cm diameter and 1.7 cm high) without feed. Newly hatched larvae about 24 h–48 h old (five days after incubation of the eggs) were collected for the bioassay.

### 2.2. Origin and Culture of Probiotics

A *P. pentosaceus* strain, KVL-B19-01, was isolated and proliferated at the Department of Plant and Environmental Sciences (PLEN), University of Copenhagen (UCPH), Frederiksberg, Denmark. The whole process of culturing, collecting, and processing this strain in freeze-dried powdered form was carried out according to the procedure described by Lecocq et al. [9]. De Man, Rogosa, and Sharpe (MRS) culture medium was used for the cultivation of *P. pentosacceus*. Bacterial cells were incubated in MRS broth at 33 °C for 24 h, collected by centrifugation (15 min at 5000 rpm), and washed thrice in phosphate buffer saline (PBS). Afterward, the cells were transferred into 50 mL Falcon tubes, centrifuged once more, and re-suspended in 20 mL PBS. All the samples were kept at −80 °C before freeze-drying. Freeze-drying was carried out over 18 h at −60 °C in a Hetosicc CD52 freeze drier (University of Copenhagen, Frederiksberg). Finally, the freeze-dried sample was preserved at −20 °C until diet supplementation. Before feed supplementation, the colony forming unit (CFU) was determined to be 1 × 10^11^ CFU·g^−1^. The *B. subtilis* strain 597 and the *E. faecium* strain 669 tested in this study were obtained from Chr. Hansen A/S, Horshølm, Denmark; *B. subtilis* 597 was provided as spray-dried with a CFU value of 5.4 × 10 ^11^ CFU·g^−1^ and *E. faecium* 669 in a freeze-dried powdered form with a CFU value of 6.8 × 10 ^11^ CFU·g^−1^.

### 2.3. Preparation of Probiotic Diet Mix

Commercially available organic oats (365 Øko^®^ Havregryn, Denmark) were used as a feed for the larvae. The control diet was composed of 25 g of plain oatmeal flour while the other three treatments consisted of the same amount of oat flour mixed with the probiotics (three bacterial strains) using a mixer (Budget^®^, Denmark). All three bacterial strains were mixed to a final concentration of 5 × 10^9^ CFU·g^−1^ in the oatmeal. Since the experiment was carried out 9 months after first collecting or producing the strains, the CFU value for *P. pentosaceus* was found to have decreased, which resulted in having to supply 1.250 g of the bacteria in the oatmeal diet; for *E. faecium* 669 and *B. subtilis* 597, 0.184 g and 0.231 g respectively were mixed in the feed. In all treatments, larvae were moved to freshly mixed oatmeal flour + bacteria at week 5 after disposing of the leftover frass. Cut potatoes were replaced in the boxes every 2 days as a source of water for the larvae.

### 2.4. Preparation of Metarhizium brunneum Inoculum

*Metarhizium brunneum* strain KVL 12 − 37 was used for the pathogen challenge experiment. The fungus was kept in the culture collection at the Department of Plant and Environmental Sciences, University of Copenhagen, Denmark, at −80 °C. Conidia were produced by culturing the fungus on Sabouraud’s 4% dextrose agar (SDA; Merck KGaA, Darmstadt, Germany) in Petri dishes for three weeks at 23 °C. Conidia were harvested by scraping the surface of the culture with a sterile loop and 0.05% Triton-X 100, and subsequently the solution was filtrated over three layers of sterile gauze to eliminate hyphae and agar. The conidia concentration was determined with a haemocytometer (Neubauer improved) and the concentration was adjusted to 2.7 × 10^6^ conidia.mL^−1^ suspension. Prior to the experiment, the viability of *M. brunneum* conidia was qualified by a germination test.

### 2.5. Bioassay Assessments

#### 2.5.1. Assessment of Larval Body Weight and Survivability of Larvae

Ten replicates of 100 larvae 24 h−48 h old were set up for each of the four treatments. In all, four thousand larvae were used for this experiment. At this age, average weight per larva was calculated by weighing 100 larvae 40 times, and the final mean weight per larva was found to be 0.00042 mg. The larvae were randomly divided and kept in plastic boxes (16.5 cm × 10 cm × 7 cm) with a vented lid. The boxes containing the larvae were incubated in darkness at 30 °C for up to 15 weeks. Weight and survival were measured weekly from week five until week fifteen or until the larvae in a replicate reached an average of 0.12 g/larva. Weight was measured using a scale with a precision of three decimal places (Sartorius Lab Equipments, Germany).

#### 2.5.2. Pathogen Challenge with *Metarhizium brunneum*

The pathogen challenge was carried out in week 9. Five larvae were randomly selected from each subgroup (i.e., 50 larvae per treatment group of which 25 larvae served as test group while the other 25 larvae served as control group). Individual larvae were weighed prior to the experiment to analyse the effect of weight on infection. The fungal challenge dose for this strain was chosen based on the LD_50_ value for this fungal strain, obtained from a study conducted by Pascal Herren, Ph.D. scholar (unpublished data) that would result in an expected 50% mortality among tested larvae. Twenty-five larvae from each treatment received the fungal treatment, with 2 µL of *M. brunneum* KVL 12 − 37 (1.22 × 10^5^ spores) conidial suspension on their integument, which was topically applied while the other twenty-five larvae were applied with 2 µL 0.05% Triton-X 100 as the control group. The larvae were transferred to 30 mL medicine cups (Hammarplast Medical AB, Sweden) lined with moist filter paper (average size 2.3 cm × 2.0 cm) attached to the inner wall to maintain humidity and kept without feed for 24 h. The next day, 2 g of plain oat flour was provided for feed and a moist filter paper for water. The cups were closed with a vented lid. The filter paper was changed daily, and the larvae were observed daily for 14 days. During the experimental period, if any dead larvae were detected, they were surface sterilized by dipping them into 5% NaOCl solution for 30 s and then three times into sterile demineralized water (DW) for 30 s. These dead larvae were finally transferred into new medicine cups with open lids and left at room temperature. The following day, a moist filter paper (average size 2.5 cm × 2.0 cm) was attached to the inner wall of the medicine cups and the paper was changed daily until mycosis was observed on the dead larvae, generally 3–5 days after death. Dead larvae were considered mycosed if the characteristic greenish sporulation of *M. brunneum* was detected on the larval surface.

### 2.6. Statistical Analysis

All statistical analyses were carried out with R version 4.0.3 [20] (The Foundation for Statistical Computing Platform, 2020). The overall effect of probiotic treatment on individual larval body weight was analysed with the nonparametric Kruskal–Wallis rank-sum test (kruskal_test function, rstatix package, R). If there was an overall significant effect (*p* < 0.05), post-hoc analyses were performed using the Dunn test with Bonferroni method of P-value adjustment for multiple comparisons (*p* < 0.05, Dunn test function, FSA package, R). The number of larvae counted weekly was analysed with a generalized linear model fitted with Poisson distribution (glm function, Poisson family, lme4 package, R). Post-hoc analyses were carried out with differences of least squares means (LS Means) with Tukey adjustment for multiple comparisons (glm function, emmeans package, R). The number of *T. molitor* larvae that survived exposure in the four treatment groups (C, B, E, and P) and two subtreatment groups (F (+): exposed to the fungus *M. brunneum*, and F (-): exposed to 0.05% Triton-X 100 as control) for a 14-day period was subjected to time-to-event analyses using a log-rank test in R (*p* < 0.05, survfit function, survival and survminer package, R). Post-hoc analysis of the log-rank test was carried out with Bonferroni adjustment. Logistic regression and odds ratio were used to analyse whether larval weight had an influence on mortality in the fungus-exposed larvae.

## 3. Results

### 3.1. In Vivo Study of Probiotic-Based Diet Supplementation Effects

Average larval weight was evaluated from week 5 onward until larvae from each treatment group were harvested at weeks 9, 13, and 15 depending on the treatment (Figure 1). At day 5, the average weight of individual larvae was 0.00042 mg. At week 9, there was an overall significant difference between treatments (χ^2^ = 36.44, d.f. = 3, *p* < 0.0001). Larvae from the *P. pentosaceus*-fed group reached an average weight of 0.133 g/larva and were significantly heavier than larvae from the *E. faecium* treatment (0.065 g/larva, *p* < 0.0001), from the control treatment (0.040 g/larva, *p* < 0.0001), and from the *B. subtilis* treatment (0.029 g/larva, *p* < 0.0001). Likewise, at week 13 there was an overall significant difference between treatments (χ^2^ = 23.28, d.f. = 2, *p* < 0.0001). At this week, larvae fed with *E. faecium* bacteria reached an average weight of 0.131 g and were significantly heavier than larvae from the *B. subtilis* treatment (0.058 g/larva, *p* < 0.0001) and from the control treatment (0.104 g/larva, *p* = 0.0021). Finally, by week 15 there was again a significant difference between treatments (χ^2^ = 14.28, d.f. = 1, *p* < 0.0001). At this week, larvae from the control treatment reached an average weight of 0.131 g/larva, which was significantly heavier than those from the *B. subtilis* treatment (0.072 g/larva, *p* < 0.0001). By counting the number of surviving larvae in each harvesting week (9, 13, and 15), the analysis did not show any significant differences in the effect of treatment groups on larval survival (*p* > 0.05). The average larval survivability in all treatment groups was above 81.07%.

### 3.2. Pathogen Challenge with the Fungus Metarhizium brunneum

Over the 14-day trial period, larval mortality in all treatment groups that were exposed to Triton-X without fungi as a control was around 10–20% and was not statistically different (*p* > 0.05). None of the unexposed dead larvae showed any visible sign of fungal infection, whereas all the dead larvae from the exposed treatments showed signs of fungal growth, with green spores developing in 75–100% of the cadavers. The study found an overall significant effect (χ^2^ = 24.70, d.f. = 3, *p* < 0.05) of the treatments on larval survival after exposure with *M. brunneum* (Figure 2). Pairwise comparison of the fungus-exposed larvae showed significant larval mortality in the *B. subtilis* treatment group compared to those from the *E. faecium* (*p* < 0.05) and *P. pentosacceus* (*p* < 0.05) treatment groups, but not from the control group (*p* > 0.05) (Figure 2.). In addition to this, there was significantly less larval mortality in the *P. pentosacceus* (*p* < 0.05) and *E. faecium* (*p* < 0.05) treatment groups compared to the control group. However, there was no difference (*p* > 0.05) in larval mortality between the *P. pentosaceus* and *E. faecium* treatment groups. (Figure 2.). Finally, the analysis did not find any significant effect of larval body weight (χ^2^= 6.69, d.f. = 3, *p* > 0.05) on larval mortality after pathogen challenge.

## 4. Discussion

The application of probiotics in the mass production of *T. molitor* is a novel approach to protection against entomopathogens in an insect colony. This concept was guided by evidence of beneficial effects of a probiotic-based diet on health and nutrition in food animals [19] and other insect models such as *Ceratitis capitata* and *Tribolium castaneum* [11,21,22]. A recent review has also presented an overview of insect–microbiota interactions and the use of probiotics in insects reared for food and feed, as well as their interactions with the host microbiota [14]. The present study investigated whether supplementing the *T. molitor* larval diet with probiotic bacterial strains, namely *P. pentosacceus* KVL-B19-01, *E. faecium* 669, and *B. subtilis* 597, could provide health and nutritional benefits to the insects. The results demonstrated that two of the probiotic strains (*P. pentosacceus* KVL-B19-01 and *E. faecium* 669) improved larval growth and disease resistance. Beneficial effects of these two strains have been documented in other studies including both vertebrate and non-vertebrate host animals [11,21,23,24,25,26]. On the other hand, this study could not demonstrate beneficial effects in either nutrition or disease resistance by supplementing the larval diet with *B. subtilis* 597, although another *B. subtilis* NCIMB 3610 strain showed improvement in growth in a study conducted by Rizou et al. [7]. In general, several studies have shown the probiotic nature of other strains of *B. subtilis* and/or its metabolites and have been routinely used for probiotic purposes in conventional livestock farming and aquaculture, as well as with honeybees and silkworms [24,27,28,29,30].

The two bacterial treatments, *P. pentosaceus* KVL-B19-01 and *E. faecium* 669, improved larval growth. These bacterial strains are classified as lactic acid bacteria (LABs) owing to their production of lactic acid as a metabolite. The improved larval growth could have been the result of modulation of gut symbionts and inhibition of pathogenic organisms after successful gut colonization. As with higher animals, insect species also depend on symbiotic relationships with microorganisms present in their gut for digestion and detoxification of harmful substances [31]. Several in vivo studies performed in mice and humans have shown successful gut establishment of lactic acid bacteria (LABs) following regular supplementation [32,33,34]. Similarly, other studies conducted in livestock animals have found successful gut colonization and nutritional benefits [35,36]. LABs colonize the gut epithelium and maintain gut health by decreasing pH, the production of bacteriocins and exo-polysaccharides, and competitive exclusion of pathogenic organisms [37,38,39]. In addition to this, studies have shown that other lactic acid-producing bacteria can stimulate host defensin production from gut epithelial cells and prevent adhesion of pathogenic microorganisms [38,39,40,41]. Moreover, antimicrobial activity of LABs against entomopathogens has been reported in vitro in studies conducted by Grau et al. [11] and Lecocq et al. [9]. Recent microbiome studies have also reported the presence of LAB strains in *T. molitor*’s gut [9,42,43]. The strain of *P. pentosaceus* that was used in this study was originally isolated from the gut of *T. molitor,* and successful gut colonization was recently shown by Lecocq et al. [9]. It is assumed that LAB strains adhered nonspecifically to the gut epithelium with the help of their positive hydrophobicity [44], and in this way modulated the gut environment to increase a healthy gut flora-favouring feed-digestion process.

Another theory behind the improved growth observed in the *P. pentosacceus* KVL-B19-01 and *E. faecium* 669 treatments could have been from contributions made to host enzymatic activity after gut colonization. Studies have reported the exogenous enzymatic activity (production of amylase, trypsin, protease, lipase, etc.) of LABs [45,46,47], and similar enzymes are secreted in the mealworm gut for feed digestion [48,49]. Thus, the synergistic effects of the probiotic strains on endogenous enzyme production could have improved digestion and absorption of nutrients in the host [47,50,51,52,53,54]. Additionally, increased host appetite is also linked to additive enzyme activity [55]. Other studies have reported that exogenous enzymatic activity of *P. pentosaceus* and *E. faecium* enhanced feed palatability and taste by production of short-chain fatty acids (SFAs) [38,56,57,58]. The increased palatability and feed intake were visually evident in this study by observing complete depletion of the diet in those treatment groups that were supplemented with *P. pentosaceus* KVL-B19-01 and *E. faecium* M74.

Furthermore, probiotic strains promote host body growth with histological modifications in the gut region [54,57,59]. It is speculated that *P. pentosaceus* KVL-B19-01 and *E. faecium* 669 might have altered midgut microvilli structure with increased permeability and absorptive area to maximize nutrient absorption by the microvilli. Previous studies have shown a similar beneficial effect of *P. pentosacceus* by increasing absorptive surface with elevated mucus-secreting goblet cells and elongated intestinal villi [25,50,60]. Likewise, it has been documented that *E. faecium* improved mucosa ultrastructure, widened the transport area, and improved intestinal permeability, thus enhancing nutrient absorption and reducing energy consumption [35,57,59]. In addition to this, the thickness of the muscular layer of the intestine is also associated with improved nutrient absorption [50].

Among the tested strains, larvae fed with *P. pentosaceus* KVL-B19-01 grew faster and first attained maximum body weight. This difference could be explained by the energy cost and extra nutrients provided by higher cell counts (either dead or alive) of the *P. pentosaceus* strain mixed in the diet. Lecocq et al. [9] found improved growth of *T. molitor* larvae by feeding with a dead/autoclaved *P. pentosaceus* strain. Similarly, other studies have also demonstrated that microbes (whether dead or alive) can serve as a rich source of protein, and dead microbes can be as effective as live ones [61,62].

The present study showed poor growth of larvae fed with *B. subtilis* 597. Though several strains of this bacterium are used as biopesticides [63], this study attempted to evaluate whether *B. subtilis* 597 can confer probiotic effects on *T. molitor* larvae. The current poor performance of this strain could be due to an unfavourable environment created by the bacterium itself. Pathogenicity possessed by this strain has been linked with the bacterial protein elicitor AMEP412, biosurfactants, broad spectrum cyclic lipopeptides, and chitinase [64,65,66,67,68]. Since insect growth and development are largely dependent on a chitinous structure, the damage to the peritrophic membrane from the chitinase enzyme can lead to a reduced nutrient utilization and consequently to poor insect growth [68]. Importantly, another reason could be the resistance possessed by *T. molitor* larvae itself against *B. subtilis*. *Tenebrio molitor* belongs to the endopterygote clade, which produces antibacterial peptides such as defensin against Gram-positive bacteria [69]. This activity has been shown in *Allomyrina dichotoma*, a member of the beetle family [70]. However, in contrast to this study, the probiotic effects of *B. subtilis* have been reported in other arthropod species such as silkworm and white shrimp [30,71,72]. More interestingly, a recent study conducted by Rizou et al. [7] found beneficial effects of *B. subtilis* on larval growth and other nutritional fortification effects in *T. molitor* larvae. This contrasting difference might be due to intraspecific differences since biochemical activity can strongly differ between bacterial strains [73].

This study found that *T. molitor* larvae fed with *P. pentosaceus* and *E. faecium* were less susceptible to *M. brunneum*, whereas higher larval mortality was observed in larvae provided with *B. subtilis*. This health-related effect of *P. pentosaceus* and *E. faecium* could be explained by the antifungal properties of these LAB strains. LABs are commonly used as fungicides, since they can synthesize organic acids (e.g., lactic acid, succinic acid, and acetic acid) and other antifungal and antibacterial metabolites such as phenylacetic acids, cyclic dipeptides, hydrogen peroxide, diacetyl, and antimicrobial peptides [74,75]. Moreover, studies have reported that gut microbiota can modulate insects’ immune response, thereby making them resilient against pathogens [75]. The antifungal activity of *P. pentosaceus* against several fungal species such as *Aspergillus fumigatus*, *Botrytis elliptica*, *Fusarium oxysporum*, *Penicillium roqueforti,* and *P. chrysogenum* has been shown [73], and for *E. faecium* against fungi such as *Penicillium* spp., *Aspergillus* spp., and *Cladosporium* spp. [18]. On the other hand, larvae fed with *B. subtilis* did not show any resistance against the fungus *M. brunneum* compared to controls. Although the results showed that larval weight did not have an effect on larval mortality after pathogen challenge, larvae fed with *B. subtilis* were visibly undernourished and began to die earlier than larvae in the *P. pentosaceus* and *E. faecium* groups after exposure to the fungus. It can be speculated that the immune function in the undernourished *B. subtilis*-fed larvae was compromised, Fitness costs in terms of resource use and other pleiotropic effects could have halted disease resistance, subsequently leading to death [76]. A recent review paper has also reported that gut microbes can assist pathogens in overcoming host immune responses [13]; that could also be assumed in this situation with higher larval mortality. Contrary to this study, other studies have shown that other *B. subtilis* strains can exhibit antifungal activity against diverse fungal species [77,78,79,80]. Thus, more studies are required to fully understand the mechanisms behind the poor performance of *B. subtilis*-fed larvae against *M. brunneum*. Finally, the isolation of the pathogen-treated larvae could be viewed critically. Since the laboratory setup differs from the industrial rearing system, this experiment cannot represent the pathogen challenge in the actual scenario as insects exhibit anti-parasitic defence mechanisms [81]. Behavioural immunity is one of the natural methods shown by insects that indirectly protects them from the surrounding pathogenic organisms [81]. Likewise, studies have reported nest hygiene maintenance in social insects such as termites and ants through cannibalism and corpse-burying behaviours that remove the dead [82]. Similar mechanisms may occur with this insect species under mass breeding conditions, which would make observation and quantification of the effects of the pathogen difficult. This study undeniably underlines that the health status of the individual insect is influenced by the gut microbiome. Thus, more studies are needed to evaluate all these effects in a mass-production setting.

## 5. Conclusions

This study demonstrated the developmental and health benefits of dietary supplementation with probiotic strains *P. pentosaceus* and *E. faecium* in *T. molitor* larvae. These two strains improved larval growth and larval resistance against the entomopathogenic fungus *M. brunneum*. However, the *B. subtilis* strain used in this study was shown to have the opposite effect on *T. molitor*, resulting in decreased larval growth and increased susceptibility to *M. brunneum*. The exact mechanism behind the probiotic effects still needs to be revealed in future studies. In addition to this, the suitability of probiotic strains for diet supplementation also requires further study.

## Figures and Tables

**Figure 1 insects-13-01114-f001:**
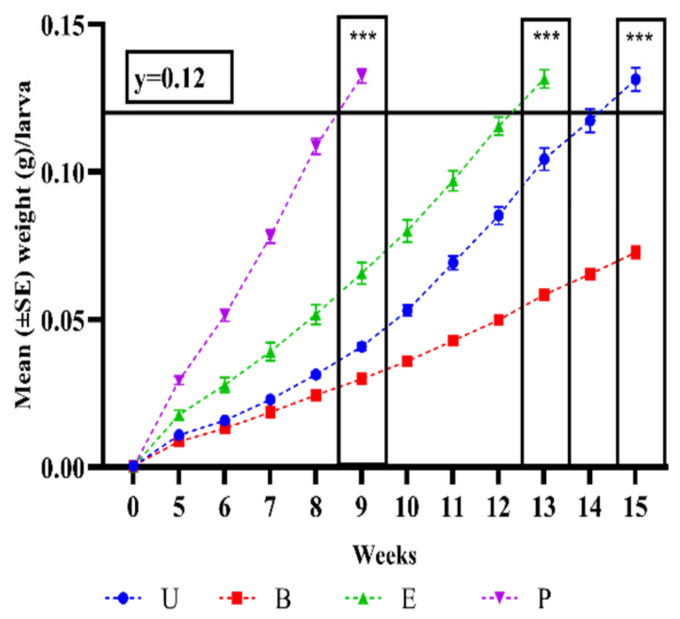
Average body weight (g) of individual *Tenebrio molitor* larvae in the four probiotic treatment groups during the experimental period of 15 weeks. Horizontal reference line at Y = 0.12 g indicates the threshold weight of the larvae to harvest. Blue: control (C); red: *Bacillus subtilis* (B); green: *Enterococcus faecium* (E); and violet: *Pediococcus pentosacceus* (P). Each treatment group consisted of 10 subgroups, each with 100 larvae. Larvae were harvested at three time points: P larvae at week 9, E larvae at week 13, and C larvae at week 15, as indicated by final data points. The bars in each curve represent the standard error (±SE) from the mean weight. Significant differences (*p* < 0.0001) in larval weight at weeks 9, 13, and 15 are indicated by ***.

**Figure 2 insects-13-01114-f002:**
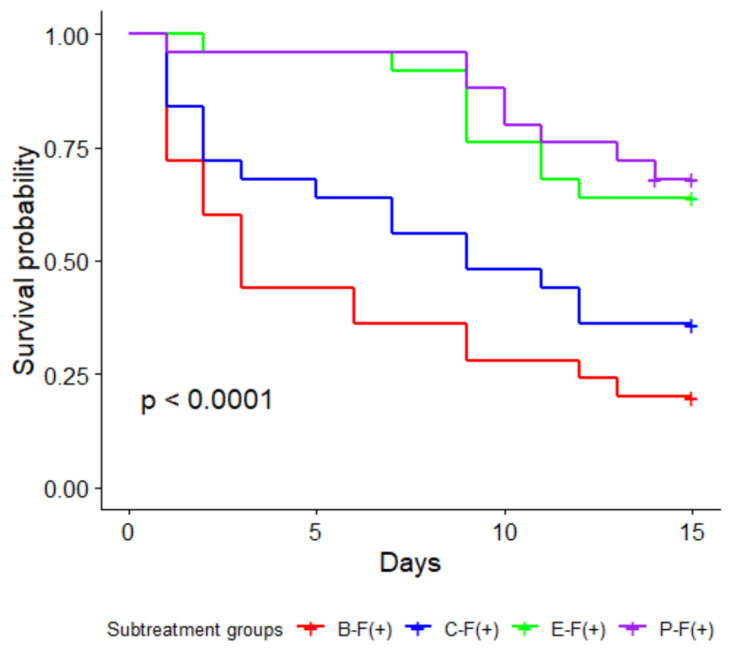
Kaplan–Meier curve showing the proportion of alive Tenebrio molitor larvae recorded over two weeks in four probiotic treatment groups. Blue: control (C); red: *Bacillus subtilis* (B); green: *Enterococcus faecium* (E); and violet: *Pediococcus pentosacceus* (P). Larvae were exposed to the fungus *Metarhizium brunneum* strain KVL 12–37 and observed for 14 days as a part of the fungal pathogen challenge experiment. Highest mortality was observed for fungus-treated larvae fed with *B. subtilis* bacterium, which was significantly different from the two probiotic treatment groups but not from the control group. The least mortality was observed for larvae supplemented with *P. pentosacceus* strains, which was significantly different from *B. subtilis* treatment groups but not from *E. faecium* and control groups. Different letters at the right end of the curves indicate significant differences (*p* < 0.05, Bonferroni adjustment for multiple comparisons) between the corresponding treatment groups.

## Data Availability

The data are available from the corresponding author upon request.

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
