# Peer review of "Effect of Probiotics on Tenebrio molitor Larval Development and Resistance against the Fungal Pathogen Metarhizium brunneum"

_insects, 2022, doi:10.3390/insects13121114_

Round 1
Reviewer 1 Report (Previous Reviewer 1)
Many thanks for the revision of the manuscript
Reviewer 2 Report (Previous Reviewer 2)
The authors have made improvements in this revision and I would like to recommend an acceptance for publication.
This manuscript is a resubmission of an earlier submission. The following is a list of the peer review reports and author responses from that submission.
Round 1
Reviewer 1 Report
Dear Authors,
Your manuscript entitled “Effect of probiotics on Tenebrio molitor larval development and resistance against the fungal pathogen Metarhizium brunneum” has been sent for my consideration.
A lot of interesting data are provided in the manuscript. However, from my point of view, some extensions in the text and also some additional data would be advantageous to improve the quality of the manuscript.
Introduction:
· Line 42-43: For completeness, please attach references on the current status of the approval for Tenebrio molitor in the EU
· Line 62-65: Please provide more recent literature to support this statement.
· Line 66-68: Can you please be more specific about the definition of probiotics for insects and to what extent one has to make differences/demarcations to the WHO definition here. The reference 21 by Savio et al that you are mentioning later on as well, deals this topic very intensively here.
· Line 71-75: Can you describe this section for insects a bit more concretely, the previous statements are rather general and not tailored to the topic of your manuscript of insect mass rearing
Material and Methods:
· Line 86: Did you evaluate the microbial load of the potato starch flour?
· Line 90: What`s the developmental stage of the larvae at day 5?
· Line 97: please insert the reference number for Lecocq et al., 2021
· See the overall chapter 2 origin and culture of probiotics: According to the terminology of both the WHO and the International Scientific Association for Probiotics and Prebiotics (ISAPP; see https://www.nature.com/articles/nrgastro.2014.66), the term "probiotics" refers to ""living microorganisms which, when administered in appropriate amounts, confer a health benefit on the host". Please comment on the viability of your cells (CFU) within this chapter in more detail.
· Line 110: Did you evaluate the microbial load of the feeding material
· Line 121: According to your results section, the larvae die as a result of inoculation with Metarhizium brunneum. But obviously the T. molitor (beetle?) seems to tolerate them. Please clarify this contradiction
· Chapter 5.2. Pathogen challenge with Metarhizium brunneum: Is the sample number of N=5 for Metarhizium brunneum sufficiently high to support benefits in mass-rearing? The controls meaning larvae posttreated with the probiotics but inviolated from Metarhizium brunneum is missing in here. In addition, the isolation of pathogen-treated larvae into 30 ml medicine cups must be viewed critically because under mass breeding conditions it is known that insects eliminate sick or dead animals by eating them in order to keep the population healthy. Please comment on this.
· Line 162: Insert the link for the software if it was used as a freely available variant or please indicate your other source of supply.
Results:
· Line 181 and Figure 1: The material and methods section states that the larvae were fed with the different diets 5 days after hatching. In the diagram 1 you give a weight of 0 mg for the larvae on day 5 after hatching - which cannot be correct because these small larvae also have a certain weight of their own. Please modify figure 1 accordingly.
· Have you recorded the feed consumption for the different feeding options? Can you make statements on FCR or other conversion parameters (ECI, FA ec. pp)? Did 100% of the animals survive the probiotics treatment?
· Chapter 3.2. Pathogen challenge with the fungus Metarhizium brunneum: In this experiment, the respective controls for the larvae that only received the probiotics are missing. Please add these and also the triton controls to figure 2.
· Line 210-211: How was it analytically proven that there was no infection. Were antifungal peptide concentrations analysed or immunohistochemical analyses performed to support this?
· Line 216 ff: why was a paired test carried out here and not a multifactorial annova?
· Line 221-222: What do you mean by "association" - is it about correlations? What does the abbreviation d.f. mean?
Discussion:
· Many thanks for the extensive discussion!
· Line 251, 270/271 and 274: Please insert the reference number
Reviewer 2 Report
Dahal et al. report the feeding of three bacterial species Pediococcus pentosacceus, Enterococcus faecium and Bacillus subtilis on the development and survival of the mealworm Tenebrio molitor over the challenge with the fungal pathogen Metarhizium brunneum. It was found that the feeding of P. pentosacceus and E. faecium but not B. subtilis could promote insect growth and defend against fungal infection. The first two bacteria thus have potential to be used as probiotics to benefit the mass rearing of this insect.
Overall, the experiment designs were simple and have not been clearly described. It has not been shown whether the feeding of these three bacteria could colonize insect guts and or cuticles. As being cited, the beneficial effect of Pediococcus pentosacceus to mealworm has already been reported by this group (J. insects as food Feed, 2021, 7, 1–15).
The authors should have been aware that the entomopathogenic fungi like M. brunneum infect insect through cuticle penetrations. It has not been cited and discussed by the authors that, actually, the ectomicrobiomes assembled on insect body surfaces may play more important roles than gut bacteria to defend against fungal parasites. The protection effect and potential mechanism of P. pentosacceus and E. faecium against M. brunneum are unclear.
It did not make any sense to use these bacteria 9 months post preparation (line 114).
Method section 5, which instar of these “24-h-old” larvae? Are they tiny just after hatching? The insects were reared up to 15 weeks while Metarhizium challenge was conducted in “week 9”, not clear about the instar/stage of these insects. When these insects had been fed with bacterial mixer?